# SEBOOST – Boosting Stochastic Learning Using Subspace Optimization Techniques

**Elad Richardson**[*1]    **Rom Herskovitz**[*1]    **Boris Ginsburg**[2]    **Michael Zibulevsky**[1]

[1]Technion, Israel Institute of Technology [2]Nvidia INC

{eladrich,mzib}@cs.technion.ac.il {fornoch,boris.ginsburg}@gmail.com

## Abstract

We present *SEBOOST*, a technique for boosting the performance of existing stochastic optimization methods. *SEBOOST* applies a secondary optimization process in the subspace spanned by the last steps and descent directions. The method was inspired by the *SESOP* optimization method, and has been adapted for the stochastic learning. It can be applied on top of any existing optimization method with no need to tweak the internal algorithm. We show that the method is able to boost the performance of different algorithms, and make them more robust to changes in their hyper-parameters. As the boosting steps of *SEBOOST* are applied between large sets of descent steps, the additional subspace optimization hardly increases the overall computational burden. We introduce hyper-parameters that control the balance between the baseline method and the secondary optimization process. The method was evaluated on several deep learning tasks, demonstrating significant improvement in performance. Video presentation is given in [15]

## 1  Introduction

Stochastic Gradient Descent (SGD) based optimization methods are widely used for many different learning problems. Given some objective function that we want to optimize, a vanilla gradient descent method would simply take some fixed step in the direction of the current gradient. In many learning problems the objective, or loss, function is averaged over the set of given training examples. In that scenario calculating the loss over the entire training set would be expensive, and is therefore approximated on a small batch, resulting in a stochastic algorithm that requires relatively few calculations per step. The simplicity and efficiency of SGD algorithms have made them a standard choice for many learning tasks, and specifically for deep learning [9, 6, 5, 10] . Although the vanilla SGD has no memory of previous steps, they are usually utilized in some way, for example using momentum [13]. Alternatively, the AdaGrad method uses the previous gradients in order to normalize each component in the new gradient adaptively [3], while the ADAM method uses them to estimate an adaptive moment [8]. In this work we utilize the knowledge of previous steps in spirit of the Sequential Subspace Optimization (*SESOP*) framework [11]. The nature of SESOP allows it to be easily merged with existing algorithms. Several such extensions were introduced over the years to different fields, such as PCD-SESOP and SSF-SESOP, showing state-of-the-art results in their matching fields [4, 17, 16].

The core idea of our method is as follows. At every outer iteration we first perform several steps of a baseline stochastic optimization algorithm which are then summed up as an inner cumulative stochastic step. Afterwards, we minimize the objective function over the affine subspace spanned by the cumulative stochastic step, several previous outer steps and optional other directions. The subspace optimization boosts the performance of the baseline algorithm, therefore our method is called the Sequential Subspace Optimization Boosting method (SEBOOST).

---

## 2 The algorithm

As our algorithm tries to find the balance between SGD and SESOP, we start by a brief review of the original algorithms, and then move to the *SEBOOST* algorithm.

### 2.1 Vanilla SGD

In many different large-scale optimization problems, applying complex optimization methods is not practical. Thus, popular optimization methods for those problems are usually based on a stochastic estimation of the gradient. Let $\min_{x \in \mathbb{R}^n} f(x)$ be some minimization problem, and let $g(x)$ be the gradient of $f(x)$. The general stochastic approach applies the following optimization rule

$$x_{k+1} = x_k - \eta g^*(x_k)$$

where $x_i$ is the result of the $i^{th}$ iteration, $\eta$ is the learning rate and $g^*(x_k)$ is an approximation of $g(x_k)$ obtained using only a small subset (mini-batch) of the training data. These stochastic descent methods have proved themselves in many different problems, specifically in the context of deep learning algorithms, providing a combination of simplicity and speed. Notice that the vanilla SGD algorithm has no memory of previous iterations. Different optimization methods which are based on SGD usually utilize the previous iterations in order to make a more informed descent process.

### 2.2 Vanilla SESOP

The *SEquential Subspace OPtimization Method* [11, 16] is an optimization technique used for large scale optimization problems. The core idea of SESOP is to perform the optimization of the objective function in the subspace spanned by the current gradient direction and a set of directions obtained from the previous optimization steps. Following the notations in Section 2.1, a subspace structure for SESOP is usually defined based on the following directions:

1. **Gradients**: Current gradient and [optionally] older ones $\{g(x_i) : i = k, k-1, \ldots k - s_1\}$
2. **Previous directions**: $\{p_i = x_i - x_{i-1} : i = k, k-1, \ldots k - s_2\}$

In the SESOP formulation the current gradient and the last step are mandatory and any other set can be used to enrich the subspace. From a theoretical point of view, one can enrich the subspace by two Nemirovsky directions: A weighted average of the previous gradients and the direction to the starting point. This will provide optimal worst case complexity of the method (see also [12].) Denoting $P_k$ as the set of directions at iteration $k$, the SESOP algorithm would solve the minimization problem

$$\alpha_k = \arg\min_{\alpha} f(x_k + P_k \alpha)$$
$$x_{k+1} = x_k + P_k \alpha_k$$

Thus SESOP reduces the optimization problem to the subspace spanned by $P_k$ at each iteration. This means that instead of solving an optimization problem in $\mathbb{R}^n$ the dimensionality of the subspace is governed by the size of $P_k$ and can be controlled.

### 2.3 The SEBOOST algorithm

As explained in Section 2.1, when dealing with large-scale optimization problems, stochastic learning methods are usually better fitted to the task then many more involved optimization methods. However, when applied correctly those methods can still be used to boost the optimization process and achieve faster convergence rates. We propose to start with some SGD algorithm as a baseline, and then apply a SESOP-like optimization method over it in an alternating manner. The subspace for the SESOP algorithm arises from the descent directions of the baseline, utilizing the previous iterations.

A description of the method is given in Algorithm 1. Note that the subset of the training data used for the secondary optimization in step 7 isn't necessarily the same as that of the baseline in step 2, as will be shown in Section 3. Also, note that in step 8 the last added direction is changed, that is done in order to incorporate the step performed by the secondary optimization into the subspace.

---
**Algorithm 1** The SEBOOST algorithm
---
1: **for** $k = 1, \ldots$ **do**
2:     Perform $\ell$ steps of baseline stochastic optimization method to get from $x_0^k$ to $x_\ell^k$
3:     Add the direction of the cumulative step $x_\ell^k - x_0^k$ to the optimization subspace $P$
4:     **if** Subspace dimension exceeded the limit: $\dim(P) > M$ **then**
5:         Remove oldest direction from the optimization subspace $P$
6:     **end if**
7:     Perform optimization over subspace P to get from $x_\ell^k$ to $x_0^{k+1}$
8:     Change the last added direction to $p = x_0^{k+1} - x_0^k$
9: **end for**
---

It is clear that *SEBOOST* offers an attractive balance between the baseline stochastic steps and the more costly subspace optimizations. Firstly, as the number $\ell$ of stochastic steps grows, the effect of subspace optimization over the result subsides, where taking $\ell \to \infty$ reduces the algorithm back to the baseline method. Secondly, the dimensionality of the subspace optimization problem is governed by the size of $P$ and can be reduced to as few parameters as desired. Notice also that as *SEBOOST* is added on top of baseline stochastic optimization method, it does not require any internal changes to be made to the original algorithm. Thus, it can be applied on top of any such method with minimal implementation cost, while potentially boosting the base method.

## 2.4   Enriching the subspace

Although the core elements of our optimization subspace are the directions of last $M - 1$ external steps and the new stochastic cumulative direction, many more elements can be added to enrich the subspace.

**Anchor points**   As only the last $(M - 1)$ directions are saved in our subspace, the subspace has knowledge only of recent history of the optimization process. The subspace might benefit from directions dependent on preceding directions as well. For example, one could think of the overall descent achieved by the algorithm $p = x_0^k - x_0^0$ as a possible direction, or the descent over the second half of the optimization process $p = x_0^k - x_0^{k/2}$.

We formulate this idea by defining *anchor points*. Anchors points are locations chosen throughout the descent process which we fix and update only rarely. For each anchor point $a_i$ the direction $p = x_0^k - a_i$ is added to the subspace. Different techniques can be chosen for setting and changing the anchors. In our formulation each point is associated with a parameter $r_i$ which describes the number of boosting steps between each update of the point. After every $r_i$ steps the corresponding point $a_i$ is initialized back to the current $x$. That way we can control the number of iterations before an anchor point becomes irrelevant and is initialized again. Algorithm 2 shows how the anchor points can be added to Algorithm 1, by incorporating it before step 7.

**Current gradient**   As in the SESOP formulation, the gradient at the current point can be added to the subspace.

**Momentum**   Similarly to the idea of momentum in SGD methods one can save a weighted average of the previous updates and add it to the optimization subspace. Denoting the current momentum as $m_k$ and the last step as $p = x_0^{k+1} - x_0^k$, the momentum is updated as $m_{k+1} = \mu \cdot m_k + p$, where $\mu$ is some hyper-parameter, as in regular SGD momentum.

---
**Algorithm 2** Controlling anchors in SEBOOST
---
1: **for** $i = 1, \ldots, \#anchors$ **do**
2:     **if** $r_i \% k == 0$ **then**
3:         Change the anchor $a_i$ to $x_\ell^k$
4:     **end if**
5:     Normalize the direction $p = x_\ell^k - a_i$ and add it to the subspace
6: **end for**
---

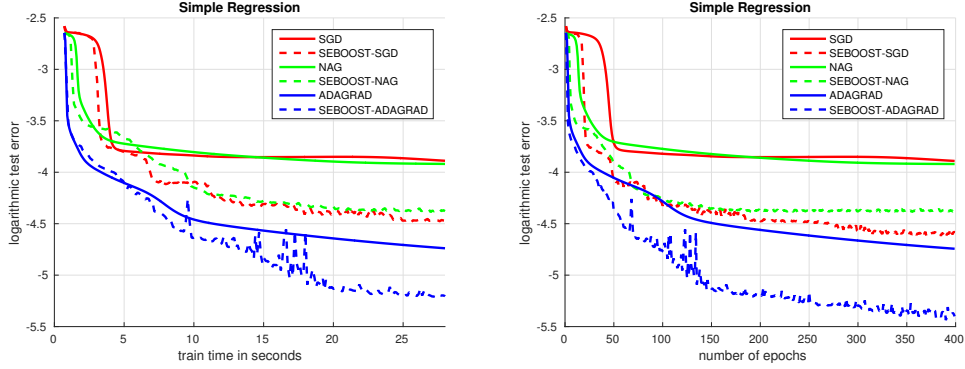

Figure 1: Results for experiment 3.1. The baseline parameters was set as $lr_{SGD} = 0.5$, $lr_{NAG} = 0.1$, $lr_{AdaGrad} = 0.05$, which provided good convergence. *SEBOOST*'s parameters were fixed at $M = 50$ and $\ell = 100$ with 50 function evaluations for the secondary optimization.

# 3 Experiments

Following the recent rise of interest in deep learning tasks we focus our evaluation on different neural networks problems. We start with a small, yet challenging, regression problem and then proceed to the known problems of the MNIST autoencoder and CIFAR-10 classifier. For each problem we compare the results of baseline stochastic methods with our boosted variants, showing that *SEBOOST* can give significant improvement over the base method. Note that the purpose of our work is not to directly compete with existing methods, but rather to show that *SEBOOST* can improve each learning method compared to its' original variant, while preserving the original qualities of these algorithms. The chosen baselines were SGD with momentum, Nesterov's Accelerated Gradient (NAG) [13] and AdaGrad [3]. The Conjugate Gradient (CG) [7] was used for the subspace optimization.

Our algorithm was implemented and evaluated using the Torch7 framework [1], and is publicly available [1]. The main hyper-parameters that were altered during the experiments were:

- $lr_{method}$ - The learning rate of a baseline $method$.
- $M$ - Maximal number of old directions.
- $\ell$ - Number of baseline steps between each subspace optimization.

For all experiments the weight decay was set at 0.0001 and the momentum was fixed at 0.9 for SGD and NAG. Unless stated otherwise, the number of function evaluations for CG was set at 20. The baseline method used a mini-batch of size 100, while the subspace optimization was applied with a mini-batch of size 1000. Note that subspace optimization is applied over a significantly larger batch. That is because while a "bad" stochastic step will be canceled by the next ones, a single secondary step has a bigger effect on the overall result and therefore requires better approximation of the gradient. As the boosting step is applied only between large sets of the base method, the added cost does not hinder the algorithm.

For each experiment a different architecture will be defined. We will use the notation $a \rightarrow_L b$ to denote a classic linear layer with $a$ inputs and $b$ outputs followed by a non-linear Tanh function. Notice that when presenting our results we show two different graphs. The right one always shows the error as a function of the number of passes of the baseline algorithms over the data (i.e. epochs), while the left one shows the error as a function of the actual processor time, taking into account the additional work required by the boosted algorithms.

## 3.1 Simple regression

We will start by evaluating our method on a small regression problem. The dataset in question is a set of 20,000 values simulating some continuous function $f : \mathbb{R}^6 \rightarrow \mathbb{R}$. The dataset was divided

into 18,000 training examples and 2,000 test examples. The problem was solved using a tiny neural network with the architecture $6 \rightarrow_L 12 \rightarrow_L 8 \rightarrow_L 4 \rightarrow_L 1$. Although the network size is very small the resulting optimization problem remains challenging and gives clear indication of *SEBOOST*'s behavior. Figure 1 shows the optimization process for the different methods. In all examples the boosted variant converged faster. Note that the different variants of *SEBOOST* behave differently, governed by the corresponding baseline.

## 3.2 MNIST autoencoder

One of the classic neural network formulation is that of an autoencoder, a network that tries to learn efficient representation for a given set of data. An autoencoder is usually composed of two parts, the encoder which takes the input and produces the compact representation and the decoder which takes the representation and tries to reconstruct the original input. In our experiment the MNIST dataset was used, with 60,000 training images of size $28 \times 28$ and 10,000 test images. The encoder was defined as three layer network with an architecture of form $784 \rightarrow_L 200 \rightarrow_L 100 \rightarrow_L 64$, with a matching decoder $64 \rightarrow_L 100 \rightarrow_L 200 \rightarrow_L 784$.

Figure 3 shows the optimization process for the autoencoder problem. A similar trend can be seen to that of experiment 3.1, *SEBOOST* is able to significantly improve SGD and NAG and shows some improvement over AdaGrad, although not as noticeable. A nice byproduct of working with an autoencoding problem is that one can visualize the quality of the reconstructions as a function of the iterations. Figure 2 shows the change in reconstructions quality for SGD and SESOP-SGD, and shows that the boosting achieved is significant in terms on the actual results.

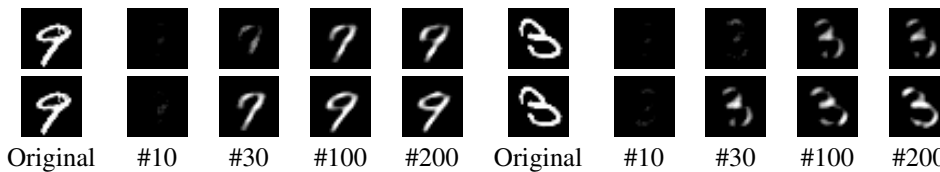

| Original | #10 | #30 | #100 | #200 | Original | #10 | #30 | #100 | #200 |

Figure 2: Reconstruction Results. The first row shows results of the SGD algorithm, while the second row shows results of SESOP-SGD. The last row gives the number of passes over the data.

## 3.3 CIFAR-10 classifier

For classification purposes a standard benchmark is the CIFAR-10 dataset. The dataset is composed of 60,000 images of size $32 \times 32$ from 10 different classes, where each class has 6,000 different images. 50,000 images are used for training and 10,000 for testing. In order to check *SEBOOST*'s ability to deal with large and modern networks the ResNet [6] architecture, winner of the ILSVRC 2015 classification task, is used.

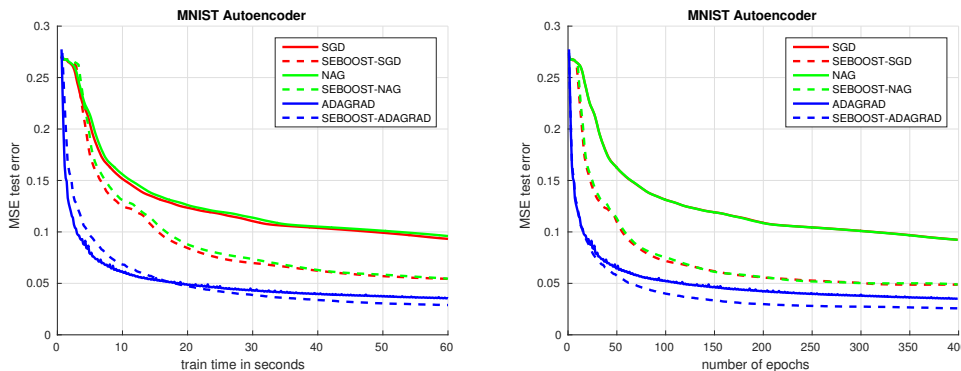

Figure 3: Results for experiment 3.2. The baseline parameters was set at $lr_{SGD} = 0.1$, $lr_{NAG} = 0.01$, $lr_{AdaGrad} = 0.01$. *SEBOOST*'s parameters were fixed at $M = 10$ and $\ell = 200$.

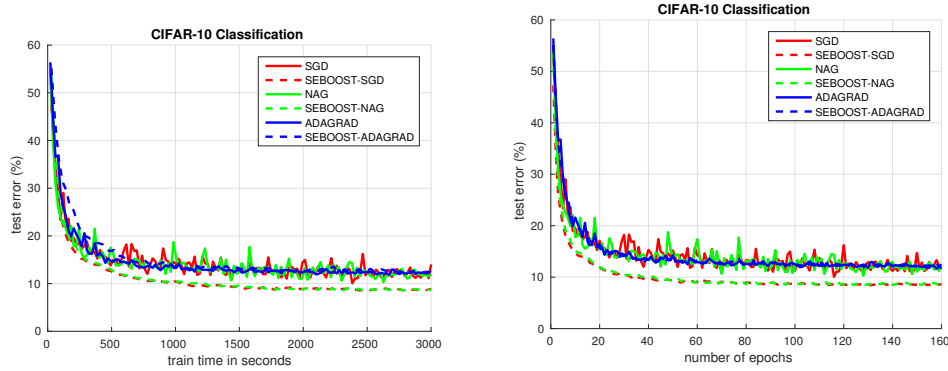

Figure 4: Results for experiment 3.3. All baselines were set with $lr = 0.1$ and a mini-batch of size 128. *SEBOOST*'s parameters were fixed at $M = 10$ and $\ell = 391$, with a mini-batch of size 1024.

Figure 4 shows the optimization process and the achieved accuracy for ResNet of depth 32. Note that we did not manually tweak the learning rate as was done in the original paper. While AdaGrad is not boosted for this experiment, SGD and NAG achieve significant boosting and reach a better minimum. The boosting step was applied only once every epoch, applying too frequent boosting steps resulted in a less stable optimization and higher minima, while applying infrequent steps also lead to higher minima. Experiment 3.4 shows similar results for MNIST and discusses them.

## 3.4 Understanding the hyper-parameters

*SEBOOST* introduces two hyper-parameters: $\ell$ the number of baseline steps between each subspace optimization and $M$ the number of old directions to use. The purpose of the following two experiments is to measure the effect of those parameters on the achieved result and to give some intuition as to their meaning. All experiments are based on the MNIST autoencoder problem defined in Section 3.2.

First, let us consider the parameter $\ell$, which controls the balance between the baseline SGD algorithm and the more involved optimization process. Taking small values of $\ell$ results in more steps of the secondary optimization process, however each direction in the subspace is then composed of fewer steps from the stochastic algorithm, making it less stable. Furthermore, recalling that our secondary optimization is more costly than regular optimization steps, applying it too often would hinder the algorithm's performance. On the other hand, taking large values of $\ell$ weakens the effect of *SEBOOST* over the baseline algorithm.

Figure 5a shows how $\ell$ affects the optimization process. One can see that applying the subspace optimization too frequently increases the algorithm's runtime and reaches an higher minimum than the other variants, as expected. Although taking a large value of $\ell$ reaches a better minimum, taking a value which is too large slows the algorithm. We can see that for this experiment taking $\ell = 200$ balances correctly the trade-offs.

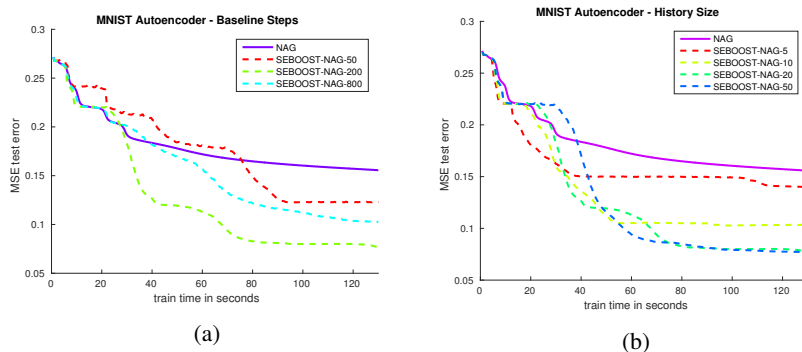

Figure 5: Experiment 3.4, analyzing different changes in *SEBOOST*'s hyper-parameters

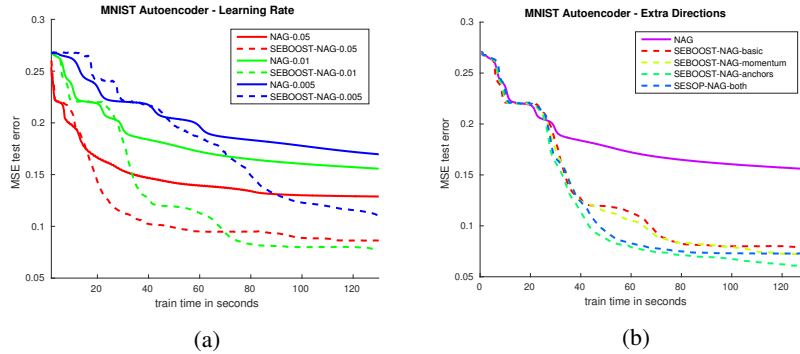

Figure 6: Experiment 3.5, analyzing different changes in *SEBOOST*'s subspace

Let us now consider the effect of $M$, which governs the size of the subspace in which the secondary optimization is applied. Although taking large values of $M$ allows us to hold more directions and apply the optimization in a larger subspace it also makes the optimization process more involved. Figure 5b shows how $M$ affects the optimization process. Interestingly, the lower $M$ is, the faster the algorithm starts descending. However, larger $M$ values tend to reach better minima. For $M = 20$ the algorithm reaches the same minimum as $M = 50$, but starts the descent process faster, making it a good choice for this experiment.

To conclude, the introduced hyper-parameters $M$ and $\ell$ affect the overall boosting effect achieved by *SEBOOST*. Both parameters incorporate different trade-offs of the optimization problem and should be considered when using the algorithm. Our own experiments show that a good initialization would be to set $\ell$ so the algorithm runs about once or twice per epoch, and to set $M$ between 10 to 20.

## 3.5 Investigating the subspace

One of the key components of *SEBOOST* is the structure of the subspace in which the optimization is applied. The purpose of the following two experiments is to see how changes in the baseline algorithm, or the addition of more directions, affect the algorithm. All experiments are based on the MNIST autoencoder problem defined in Section 3.2.

In the basic formulation of *SEBOOST* the subspace is composed only from the directions of the baseline algorithm. In Section 3.2 we saw how choosing different baselines affect the algorithm. Another experiment of interest is to see how our algorithm is influenced by changes in the hyper-parameters of the baseline algorithm. Figure 6a shows the effect of the learning rate over the baseline algorithms and their boosted variants. It can be seen that the change in the original baseline affects our algorithm, however the impact is noticeably smaller, showing that the algorithm has some robustness to the original learning rate.

In Section 2.4 a set of additional directions which can be added to the subspace were defined, these directions can possibly enrich the subspace and improve the optimization process. Figure 6b shows the influence of those directions on the overall result. In SEBOOST-anchors a set of anchor points were added with the $r$ values of $500, 250, 100, 50$ and $20$. In SEBOOST-momnetum a momentum vector with $\mu = 0.9$ was used. It can be seen that using the proposed anchor directions can significantly boost the algorithm. The momentum direction is less useful, giving a small boost on its own and actually slightly hinders the performance when used in conjunction with the anchor directions.

## 4 Conclusion

In this paper we presented *SEBOOST*, a technique for boosting stochastic learning algorithms via a secondary optimization process. The secondary optimization is applied in the subspace spanned by the preceding descent steps, which can be further extended with additional directions. We evaluated *SEBOOST* on different deep learning tasks, showing the achieved results of our methods compared to their original baselines. We believe that the flexibility of *SEBOOST* could make it useful for different learning tasks. One can easily change the frequency of the secondary optimization step, ranging from

frequent and more risky steps, to the more stable one step per epoch. Changing the baseline algorithm and the structure of the subspace allows us to further alter *SEBOOST*'s behavior.

Although this is not the focus of our work, an interesting research direction for *SEBOOST* is that of parallel computing. Similarly to [2, 14], one can look at a framework composed of a single master and a set of workers, where each worker optimizes a local model and the master saves a global set of parameters which is based on the workers. Inspired by *SEBOOST*, one can take the descent directions from each of the workers and apply a subspace optimization in the spanned subspace, allowing the master to take a more efficient step based on information from each of its workers.

Another interesting direction for future work is the investigation of pruning techniques. In our work, when the subspace if fully occupied the oldest direction is simply removed. One might consider more advanced pruning techniques, such as eliminating the direction which contributed the least for the secondary optimization step, or even randomly removing one of the subspace directions. A good pruning technique can potentially have a significant effect on the overall result. These two ideas will be further researched in future work. Overall, we believe *SEBOOST* provides a promising balance between popular stochastic descent methods and more involved optimization techniques.

**Acknowledgements**

The research leading to these results has received funding from the European Research Council under European Unions Seventh Framework Program, ERC Grant agreement no. 320649 and was supported by the Intel Collaborative Research Institute for Computational Intelligence (ICRI-CI).

## Footnotes

[1]https://github.com/eladrich/seboost

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
