[Reviews · NeurIPS 2016]

Reviewer 1

Summary

The paper proposes to improve stochastic gradient algorithms by periodically applying an additional optimization in a subspace spanned by a set of previously stored then-locally-optimal directions. It is an adaptation of the Sequential Subspace Optimization (SESOP) framework of Nakiss & Zibulevsky to the minibatch setting. Experiments show that it can improve the performance compared to the baselie stochastic gradient descent algorithms that it is used in conjunction with.

Qualitative Assessment

The paper is overall clearly written, but one important aspect of the algorithm remains not sufficiently expounded: how precisely the subspace optimization is carried over. The paper only mentions in passing that it uses conjugate gradient (CG), but a number of points would deserve further clarification: a) is CG done over a *single* larger minibatch? And how precisely is this minibatch chosen. b) specify how you compute/provide the gradients needed for that CG subspace optimization, and what is the associated computational cost c) that optimization is still not convex, so you must be meaning nonlinear-CG? Which version/implementation do you use? The computational cost *and* additional memory requirement (as this can constitute a practical limitation for large nets) for the subspace optimization would need to be disclosed and made precise. Since in effect multiple CG optimizations are performed on different minibatches, it would have been welcome to contrast this approach, at least in discussion, with attempts to adapt CG to the stochastic/minibatch case. Experimental results convincingly show that the approach can give a boost to minibatch-SGD-with-momentum, Nesterov accelerated gradient, and adagrad. But for vanilla SGD at least I would have appreciated to see the use of a learning rate decay rather than a fixed learning rate, and the assurance that all optimization hyperparameters (starting learning rate, momentum, learning rate decay) had been properly hyper-optimized to yield the lowest training error after the number of considered and displayed epochs. Captions in the figure stating that "baseline parameters was set .... which provided good convergence" is not precise enough (what was your criterion for "good convergence" and how extensively were these parameters tuned). Also why, for an optimization method, do you report only test error in your graphs (all graphs are labeled with "test error")? Wouldn't the average error over the whole *training* set have been more appropriate, as it wouldn't risk conflating optimization and generalization ability? The best would be to show both. In line 156 you comment Figure 3 saying "SEBOOST is able to significantly improve SGD and NAG and shows improvement over AdaGrad although not as noticeable", but I see exactly the contrary happening in Figure 3 (unnoticeable difference for SGD and NAG)!?! Your statement regarding SGD qualitatively supported by Figure 2 but then this is contradicted by Figure 3... Further detailed remarks: l 82:"as few parameters as required" plase clarify l 103: "current gradient": specify what is meant precisely by current gradient l 107 "useful to normalize": how precisely, and why is it useful to normalize a *direction*?

Confidence in this Review

3-Expert (read the paper in detail, know the area, quite certain of my opinion)


Reviewer 2

Summary

Authors investigate combining stochastic minimization with subspace optimization for training deep networks.

Qualitative Assessment

---------- Technical Quality ---------- The use of ResNet is commendable. However, CIFAR is a small dataset, which obscures the relative cost of (over the entire dataset) CG solution versus the cost of stochastic minimization. The most compelling demonstration would be an improvement on a large dataset with a realistic architecture. ---------- Practical Impact ---------- It is unfortunately rare in deep learning that optimization algorithms improve over heavy-ball-style momentum. RMSProp/ADAM/``nonstationary AdaGrad'' is pretty much the only widely adopted advance in the last decade. Specifically for SEBOOST, it is unclear whether it works well with stochastic regularizers (dropout/zoneout/swapout/etc.) or saddle point ``gradient reversal'' optimization (GANs/domain adaption/transfer learning/counterfactual estimation/etc.). Nonetheless, the slow convergence to the final solution of stochastic methods is annoying. Practitioners mess around with a combination of 1) learning rate schedules, 2) gradient batch size schedules, and 3) momentum schedules in order to alleviate this problem. These are effective but the main issue is the burden of hyperparameter (hyperschedule) tuning. If SEBOOST could demonstrate either 1) hyperparameter robustness or 2) effective heuristics for determining hyperparameter schedules (adaptively during training?); then this would garner interest. The good performance on convnets begs the question whether, for convnets, initially optimizing with SGD and then finishing off with a quasi-Newton method (L-BFGS/L-SR1 trust region/etc.) is an effective strategy. -------- Other -------- Authors are to be commended for presenting graphs with test error, rather than training error; too many papers promoting an optimization scheme for machine learning fail to ask whether training loss improvements are meaningful. Nonetheless, it would be helpful to view the training loss and outer iterate change magnitude as well. Consider Figure 4: the curves are consistent with the latest understanding of stochastic gradient methods on non-convex problems: that with a fixed learning rate they eventually settle into a random walk inside a sublevel set around a local optima. SEBOOST might beneficially exploit the random walk in this regime via periodic high-precision CG solution over this ``random walk ball''; this could be seen as stable iterates for SEBOOST relative to SGD and NAG. Meanwhile, Adagrad will also dampen the random walk as the effective learning rate goes to zero, which might be counterproductive for constructing the basis for the CG step. This would be seen as a stabilized iterates but worse training loss for (unboosted) Adagrad relative to (unboosted) SGD and NAG. Intuitively it seems wasteful to do the subspace optimizations until SGD has hit the ``random walk'' regime; the hallmark of this would be that the old directions would have small angles between each other in the regime of ``gradient consistency''. Monitoring such a quantity might allow one to adapt M while training, starting with M=1.

Confidence in this Review

2-Confident (read it all; understood it all reasonably well)


Reviewer 3

Summary

This paper propose an algorithm, called SEBOOST, that improves the performance of learning algorithms and optimization techniques. The algorithm is applied on top of any optimization techniques. More specifically, after running several iterations of the original optimization technique, the SEBOOST technique is applied to explore the subspace spanned by the last steps and descent directions to improve the performance. The authors propose different variations and through numerical simulations show that the SEBOOST technique can significantly improve the performance of three investigated optimization techniques (SGD, NAG and AdaGrad).

Qualitative Assessment

The paper is well-written, easy to understand and the ideas were clearly explained. The fact that the algorithm can be applied on top of standard optimization techniques makes it very appealing from practical point of view. I think comparison with other boosting techniques should significantly improve the quality of the paper and would be really interesting to see if authors have some preliminary results to discuss. Also, the fact that AdaGrad was not significantly improved in the CIFAR-10 experiment makes me curious to know how does the algorithm compares with state-of-the-art optimization algorithms in different classification/regression tasks. But overall, I think the proposed approach is very interesting and I am looking forward to the public release of its implementation. This is a big plus in my opinion :) A minor comment: I think the font-size for the figures can be enlarged a little bit to make it easier to read.

Confidence in this Review

2-Confident (read it all; understood it all reasonably well)


Reviewer 4

Summary

SEBOOST is proposed in this work, which is a technique for boosting stochastic learning algorithms via a secondary optimization process. The secondary optimization is applied in the subspace spanned by the receding descent steps, and they can be further extended with additional directions. Case studies are carried out including simple regression, MNIST autoencoder, and CIFAR-10 classifier. The performance is improved with the application of SEBOOST.

Qualitative Assessment

With a secondary optimization process, SEBOOST is proposed to boost stochastic learning algorithms. The secondary optimization process is trying to get a balance between stochastic gradient descent (SGD) and sequential subspace optimization method (SESOP), via two hyper-parameters. Case studies show that the performance is better with SEBOOST. (1) Results in Figs. 3 and 4 show that, SEBOOST works better with AdaGrad in experiment 3.2, while it works better with NAG in experiment 3.3, could the authors explain the reason? Is there any relation with the selection of the hyper-parameters? (2) This is no definition for ADAM in line 25. (3) For Figs. 3-4, abbreviations ‘ADAGRAD’ is different from that in line 157 ‘AdaGrad’.

Confidence in this Review

2-Confident (read it all; understood it all reasonably well)


Reviewer 5

Summary

The authors propose to boost the performance of stochastic gradient algorithms by incorporating an optimization problem along a subspace defined by current and previous gradient vectors and/or increments of the iterates. In one implementation, several steps of the stochastic gradient algorithm are performed, followed by a boosting step, and then back to the stochastic gradient method, and so forth. The construction is illustrated by simulations. Related ideas appear in the adaptive filtering literature where, for example, least-mean-squares and recursive least-squares methods can be intertwined to speed up convergence. The article would have benefited from some analysis and performance guarantees.

Qualitative Assessment

The authors propose to boost the performance of stochastic gradient algorithms by incorporating an optimization problem along a subspace defined by current and previous gradient vectors and/or increments of the iterates. In one implementation, several steps of the stochastic gradient algorithm are performed, followed by a boosting step, and then back to the stochastic gradient method, and so forth. The construction is illustrated by simulations. The article would benefit from some analysis and performance guarantees.

Confidence in this Review

2-Confident (read it all; understood it all reasonably well)


Reviewer 6

Summary

This paper proposes to use sequential subspace optimization framework to boost existing stochastic optimization methods, which introduces soma parameters to balance between the baseline stochastic steps and subspace optimization. Experiments show some improvement over the baseline optimization methods, e.g., SGD, NAG.

Qualitative Assessment

The technical depth and novelty is not sufficient. First, the Sequential subspace optimization method has been already proposed and applied in large scale optimization, which includes the scenario of stochastic optimization. Second, it lacks theoretical results/analysis on how much the Sequential subspace optimization method can improve, for example, in terms of convergence rate for SGD. This paper itself does not provide novel design in the boosting technique itself. The experiments do not provide comparison on other boosting techniques.

Confidence in this Review

2-Confident (read it all; understood it all reasonably well)